# Comparing Dashboard and Virtual Reality Wizard-of-Oz Setups In a Human-Robot Conversational Task

Jūra Miniota*
KTH Royal Institute of Technology
Stockholm, Sweden
jura@kth.se

Ekaterina Torubarova*
KTH Royal Institute of Technology
Stockholm, Sweden
ekator@kth.se

André Pereira
KTH Royal Institute of Technology
Stockholm, Sweden
atap@kth.se

## ABSTRACT

This paper explores the potential of using Virtual Reality (VR) technology to teleoperate social robots. A novel Wizard-of-Oz (WoZ) system using a VR headset is compared to a more traditional dashboard system in a conversational task. Participants held two conversations with a Furhat robot operated by each of the WoZ systems. Our pilot study evaluates how these two methods of operating social robots may affect the user's perception of the interaction. Results show that while both systems were perceived as autonomous, the VR system was preferred by the participants. It was rated as more enjoyable and socially immersive, and the robot's features were rated as more appropriate. Our findings suggest that VR-mediated WoZ systems can provide highly naturalistic and spontaneous data in human-robot communicative settings.

## CCS CONCEPTS

• **Human-centered computing → Mixed / augmented reality**; **User studies**.

## KEYWORDS

VR, XR, Wizard-of-Oz, Teleoperation, Social Robotics

**ACM Reference Format:**
Jūra Miniota, Ekaterina Torubarova, and André Pereira. 2023. Comparing Dashboard and Virtual Reality Wizard-of-Oz Setups In a Human-Robot Conversational Task. In *Proceedings of Virtual, Augmented, and Mixed-Reality for Human-Robot Interactions at HRI 2023 (VAM-HRI '23)*. ACM, New York, NY, USA, 8 pages. https://doi.org/XXXXXXX.XXXXXXX

## 1 INTRODUCTION

The Wizard-of-Oz (WoZ) methodology has become a popular tool for modeling naturalistic human-robot interactions in a well-controlled manner. The WoZ approach involves an operator remotely controlling the robot's behavior, e.g. the utterances it produces or the gestures it performs. In a WoZ study, the participants are usually unaware that the robot they are interacting with is controlled by a human and believe the system is autonomous, only being debriefed after the experience. Despite its widespread usage, the role of the

*Both authors contributed equally to this research

WoZ operator is often overlooked and the effect that the choice of teleoperation method may have on the interaction is rarely considered. This paper will compare two WoZ systems, a typical dashboard GUI and a virtual reality (VR) setup using a VR headset, to explore how the choice of system can affect the interaction.

The development of social robots for human-robot interaction has long been a goal of the robotics research community. The ability to interact socially with humans depends on the robot's multimodal social perception of the environment, as well as its ability to generate appropriate behavior [3, 15]. In conversational settings, humans pay attention to non-verbal cues in order to determine how they should proceed in the interaction [8]. Although robots have yet to match the performance of humans in perceiving natural language and non-verbal behavior, and producing believable multimodal responses, significant advances have been made through the use of Wizard-of-Oz (WoZ) interfaces to collect data and train machine learning models [9]. Furthermore, recent developments in perception tasks such as speech recognition [2] and object recognition, in combination with advances in generative technology [13], have the potential to enable highly complex social behavior in robots.

Using WoZ systems, it is important to argue the potential feasibility of the implementation of an autonomous system with similar behavior [18]. Recent advances in machine learning have enabled more expressive implementations of autonomous behavior. Consequently, new ways of collecting and controlling data using a WoZ setup are becoming increasingly feasible.

In this paper, we suggest a virtual reality (VR) teleoperation setup for a conversational task using the Furhat robot, in which the operator can control facial expressions, head and eye movements, and deliver utterances with lip sync in real-time. We assume that this setup can improve the user experience in human-robot interaction (HRI) studies, as it brings the conversation to a more naturalistic level. However, it can also cause the 'uncanny valley' effect, resulting in a worse perception of the robot [? ]. To test our system, we conducted an exploratory pilot user study, comparing the dashboard GUI and VR setup in a conversational setting, aiming to examine the user perception of the two setups. We hypothesize that in a conversational setting, the VR WoZ system can lead to richer data than a dashboard setup, by providing a more naturalistic interaction.

## 2 RELATED WORK

### 2.1 Dashboard WoZ

Typically, a WoZ interface involves a 'control panel' with buttons referring to the robot's action [18, 19]. This straightforward way of operating the robot is easy to use, even for a novice user. However,

manually operating the robot's behavior using a dashboard imposes certain limitations. Increasing the behavioral options on the dashboard increases the cognitive load on the operator, potentially leading to errors and delays in selecting an appropriate response. For instance, [23] report that their prototypical dashboard interface became cluttered with options of responses, which led to manual errors in clicking on the response boxes.

In a brain imaging study [17] the authors investigated how conversational dynamics vary between human-human and human-robot wizarded conversation. They found that the participants displayed more humor and spoke faster with more prosodic variation towards the human than the robot. The authors attributed this difference to the general differences in the agent's nature perception. However, the data also showed that, for the same duration of the conversation, the human agent produced more intra-pausal units and tokens than the wizarded robot agent, while the pauses and gaps were longer for the robot agent [16].

The way to avoid these drawbacks can be to decrease the number of dashboard options, which limits the interaction; or create a complex behavioral tree, which may be rather costly for each new study. Another solution would be to have an option for the operator to type an arbitrary response or have a verbal input translated into text, however, this may pose a challenge in terms of the accuracy of speech-to-text translation, as well as increased delays in response. A potential solution for reducing the operator's workload was proposed in [12], where the setup in a navigation task consisted of two wizards: a dialogue manager and a robot navigator. This approach could be further explored in order to increase the efficiency of the operator-machine interaction.

Another limitation of a dashboard WoZ system is that commonly the operator has a third-person perspective on the interaction by viewing the interaction from an external camera. A third-person perspective is inherently different from the first-person perspective [20], as the former does not provide the operator with an increased sense of presence that the latter does, which gives them a better understanding of the social situation [14]. This difference in perspective may be crucial in certain interactive settings, e.g. in a multi-party setting, when the robot's eye and head movements and posture need to be constantly adjusted to convey the desired social meaning [14, 20].

## 2.2 Virtual Reality WoZ

VR teleoperation involves an operator controlling the robot by wearing a VR headset, often for medical or industrial applications such as grasping, navigation, and object manipulation [11]. In this work, we explore the use of VR to control the robot's social features in order to model a naturalistic human-robot interaction. The operator wearing a VR headset is able to produce utterances and facial expressions that are displayed by the robot. The advantages of this VR WoZ setup, as compared to the dashboard setup, are the unconstrained variety of responses that can be produced, the absence of delay caused by manual operation, and the first-person perspective on the interaction. Moreover, a VR teleoperation platform allows for data collection of spontaneous situated behavior produced by the teleoperator without the need for intrusive sensors on the participant of the experiment [24].

However, there are issues with teleoperation systems created with the intent of deceiving participants into believing the robot is autonomous. A primary problem with VR-controlled conversational robots is that the human-like speech and behavior of the robot may disrupt the interaction if the participant begins to doubt the robot's proclaimed autonomy during the experiment. Additionally, the level of expressiveness and consistency of the VR WoZ setup depends on the operator, as they have to always produce appropriate facial expressions and utterances in real-time.

A VR-mediated WoZ system was implemented in [14], where the participants operated the robot either alone or divided the task among two operators. In this experiment, the robot served food to a customer played by an actor. The teleoperation included controlling the robot's verbal and non-verbal behavior. The introduced system suggested a compromise between solo or paired operation modes depending on the task complexity. This paper provides valuable insight into the operator's experience in terms of immersion, engagement, the ability to observe the customer, and other features. However, the human perception of the interaction (the robot's behavior) could not be evaluated since the role of the customer was played by a single actor.

Mapping the facial features of the operator onto the robot Furhat has been accomplished in a previous study [7]. However, that prior study did not involve the operator being immersed in the robot's perspective. To the best of our knowledge, the implementation of a VR teleoperation system that affords this level of control and immersion within a social setting has still not been investigated.

## 2.3 Comparing Different WoZ Systems

Overall, a direct comparison of different teleoperation methods is lacking in the HRI literature. In the paper by [22], the authors suggested a VR setup for controlling the Pepper robot as an alternative to a dashboard setup. In this study, the operator controlled only the head and arm movements of the robot. Also, the operators only tested the VR system without comparing it to the dashboard setup for the same task; and the interaction with the users was not implemented.

## 3 SYSTEM

### 3.1 Social Robot

In this work, we used the humanoid robotic head Furhat [1], capable of advanced speech synthesis and speech recognition, as well as precise facial features with a high demand for lip sync (Fig. 1). The robot's face is back-projected on the plastic mask, allowing for a wide range of facial expressions, as well as different appearances. It is equipped with a large voice library, comprising both male and female voices. Additionally, the Furhat robot is capable of displaying believable eye-gaze and performing realistic facial movements (for lip-synchronization and expressions that reveal emotional states such as sadness and excitement).

### 3.2 Dashboard

For the dashboard setup, we used a semi-autonomous dialogue generation system developed in [10]. This system was designed as a Finite-State Automaton with pre-written utterances related to casual conversational topics such as personal introductions, work,

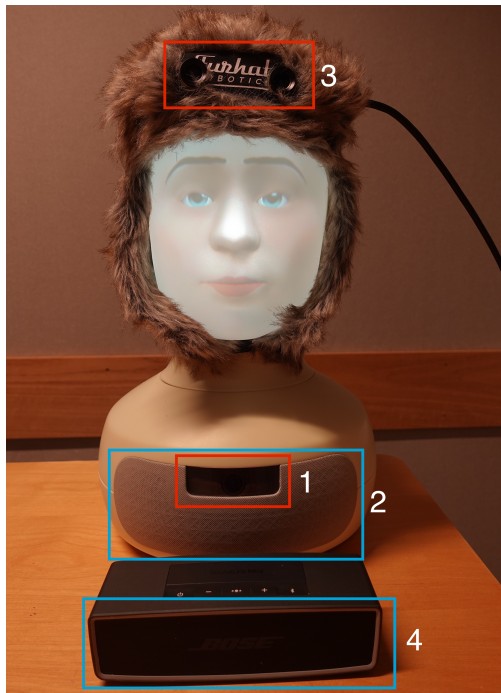

**Figure 1: Furhat robot. 1) Built-in static camera 2) Built-in speaker 3) Stereo camera 4) External speaker**

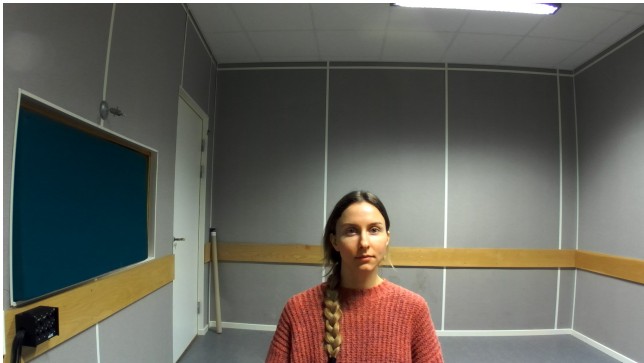

**Figure 2: View of the participant from the stereo camera.**

favorite sports, or family. To evaluate the system, it was implemented in a robot-mediated language practice cafe for second language learners [4]. The original dialogue was in Swedish; however, for this experiment, a subset of dialogue topics was selected and translated into English.

The robot initiated the conversation by greeting a participant. Upon the initiation of the conversation, the operator was presented with a list of dynamically updated utterances on the screen (Fig. 4). Then, the desired following utterance could be chosen by pressing the corresponding key on a keyboard. In the current setting, a maximum of 14 options were displayed at a time, out of which 7 predefined options were always present ("Yes", "No", "I'm not sure", "Oh, really?", "I don't remember", "I'm not allowed to tell you", "Thank you for talking to me, Goodbye"). Once the conversation tree was exhausted, the operator could suggest switching to another topic or to end the interaction. The operator could press the space key to repeat an utterance. In our experiment, the operator followed the same order of topics: work, sports, hobbies, and family.

It is worth noting that the dashboard setup in this study has some aspects of semi-autonomy compared to more basic dashboard setups. In a simple dashboard setup, all possible options are present for the operator at the same time. In the setup used in this study, the cognitive load on the operator is significantly reduced by limiting the choice of utterances.

The robot could attend to the active user by automatically moving its head and eyes, employing built-in face detection. Furthermore, Furhat was programmed to employ its head movements in

interaction through the integration of built-in gestures into its utterances, such as:

*Welcome! <gesture.Nod> My name is Furhat. What's your name?*
*What do you like to do in your spare time? <gesture.Blink>*
*Is there anything else you enjoy doing? <gesture.BrowRaise>*

### 3.3 VR Headset

We created a telepresence platform for the Furhat robot using the Meta Quest Pro[1], a VR headset that has built-in eye and face tracking hardware and software capabilities. The facial expressions, head, and eye movements of the user wearing the headset were collected from the headset and displayed on the Furhat robot's mask in real-time (Fig. 3). We used Meta platforms Movement SDK for Unity[2] and a Furhat CSharp interface[3] inside the Unity[4] Editor to integrate the VR headset with the Furhat robot to create a VR-mediated WoZ interface.

While Furhat is equipped with a built-in camera at its base, it was not suitable for the VR setup. It was fixed, thus not allowing for a full immersion for the operator; and it was located significantly below eye level. Instead, we used a stereo camera inserted into the robot's hat. This camera was located slightly above the eye level of the robot but oriented in a way that the operator could keep the head straight (Fig. 1). This way, the robot looked straight into the participant's eyes when the operator did so. It also allowed the operator to move their head freely, maintaining a first-person view (Fig. 2). The camera was directly connected to the computer to reduce lag and displayed the video to the operator in real-time.

## 4 METHOD

### 4.1 Participants

In the pilot study, 6 participants (aged 20 to 30, 5 male and 1 female) took part. All of the participants were fluent in English and two were native speakers of Swedish, one was a native speaker of English (UK), and three were native speakers of Russian. Additionally, all of the participants were either studying or working in the tech field. Of the six participants, five reported having no prior interaction with

---

[1]https://www.meta.com/se/en/quest/quest-pro/
[2]https://developer.oculus.com/documentation/unity/move-overview/
[3]https://github.com/andre-pereira/FurhatCSharpInterface
[4]https://unity.com/

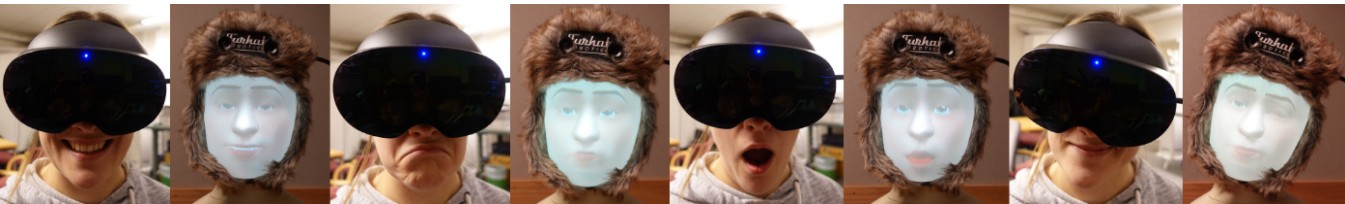

**Figure 3: Face tracking with a VR headset displayed on Furhat's face mask showing the operator's face and the corresponding facial expression on the robot.**

**Figure 4: The dashboard interface in the middle of a question flow. 1) Predefined always present options 2) Dynamic options updating along the conversation 3) Corresponding keys on the keyboard**

robots, and one participant had had a conversational interaction with a social robot. As a reward for taking part in the study, the participants were given a 50 SEK voucher. Furthermore, all of the participants gave written consent to participate in the study.

### 4.2 Operator

The operator (one of the authors) was a 27 years old female PhD student in social robotics, a native speaker of Swedish and Lithuanian with high English proficiency. The operator had a good understanding of how both interfaces work and practiced with them prior to the experiments to ensure consistent quality throughout. In both setups, the operator wore glasses.

### 4.3 Experimental Setup

The experiment was conducted in two adjacent rooms: the control room and the experimental room. The experimental room was soundproof with no visual connection to the control room. The robot was installed in the experimental room, and the participants were seated in front of the robot at the same eye level.

In the VR condition, the operator's voice was played through an external speaker placed in front of the robot. In the dashboard condition, the robot's voice was played through the built-in Furhat speaker. The participant's and the robot's voices were recorded with a high-quality USB microphone and passed to the operator seated in the control room via headphones. This meant that the operator could hear both the participant and the robot in real-time.

The robot's voice was a male state-of-the-art text-to-speech generated voice in the dashboard condition, and a female human voice in the VR condition (the operator's natural voice). The same gender-neutral mask was used for the robot in both conditions.

In both conditions, the operator could see the participant through the hat camera (in the dashboard condition - on a separate monitor).

The interaction was recorded with the microphone mentioned above and three cameras: one camera facing the participant, another facing the robot, and the third camera facing the operator.

### 4.4 Procedure

The participants were invited to have a conversation with a social robot. They were instructed that after the first conversation, the system will be changed and they will have another conversation with the same robot but with a different system in it. The order of the conditions was randomized between the participants.

First, the operator initiated the conversation. After finishing it, the participants filled in the survey about the interaction and the robot's features. Following, they were instructed to have a conversation with the new system. They were informed that the robot's questions could repeat because the new system has no memory of the first interaction. Then the participants had the second conversation, after which they filled in a survey about the second interaction and the robot's features, as well as 3 open-ended questions comparing the two systems.

In the VR condition, the operator followed roughly the same script as in the dashboard condition, asking questions about work, favorite sports, family, and where that participant lives. However, it should be noted that the operator had the freedom to say anything and take the conversation in any direction, allowing for a more natural, conversational flow. Both conversations lasted from 2.5 to 3.5 minutes.

We did not claim anything about the robot's autonomy to the participants prior to or during the experiment. Before starting the experiment, the participants did not meet the operator. After the experiment, the participants were debriefed and introduced to the operator.

| Domain | Question | Scale | Paper |
|---|---|---|---|
| Engagement | How engaging was the interaction? | 1 | Sidner et al. [21] |
| | How engaged did you think the robot was with you? | 1 | Hall et al. [5] |
| | How completely were your senses engaged in the interaction? | 1 | Sidner et al. [21] |
| | The experience caused real feelings and emotions for me. | 2 | Sidner et al. [21] |
| | I was so involved in the interaction that I lost track of time. | 2 | Sidner et al. [21] |
| Enjoyment | I enjoyed it when the robot was talking to me. | 2 | Heerink et al. [6] |
| | I enjoyed talking with the robot. | 2 | Heerink et al. [6] |
| | I found the robot enjoyable. | 2 | Heerink et al. [6] |
| | I found the robot interesting. | 2 | Heerink et al. [6] |
| Social Presence | When talking with the robot, I felt like talking with a real person. | 2 | Heerink et al. [6] |
| | I occasionally felt like the robot was actually looking at me. | 2 | Heerink et al. [6] |
| | I often realized the robot was not a real person. | 2 | Heerink et al. [6] |
| | Sometimes it seemed as if the robot had real feelings. | 2 | Heerink et al. [6] |
| Robot's Features | How appropriate were the robot's facial expressions? | 1 | — |
| | How appropriate was the robot's voice? | 1 | — |
| | How appropriate was the content of the robot's utterances? | 1 | — |
| | How appropriate was the content of the robot's utterances? | 1 | — |
| | How appropriate were the robot's reactions to what you were saying? | 1 | — |

Table 1: Interaction and robot's features questionnaires. Scale 1: 1 = not at all, 5 = very much. Scale 2: 1 = strongly disagree, 5 = strongly agree

## 4.5 Questionnaires

After each interaction, the participant filled out a questionnaire that evaluates the interaction and the robot's features. For the interaction part of the questionnaire, we compiled the questions from existing questionnaires from HRI and HAI literature. We evaluated the interactions in terms of social engagement, enjoyment, and social presence using 5-point Likert scale questions (see Table 1). The original questions were adjusted when necessary to better fit the interaction, e.g. 'When working with the robot...' was changed to 'when talking to the robot...'. The other section, designed by the authors, comprised the questions regarding the robot's features that were manipulated differently between the conditions (see Table 1).

Additionally, after finishing both interactions, we asked the following open-ended questions comparing the two systems and evaluating the participants' belief in the robot's autonomy:

(1) Did you feel any differences between the systems? If yes, what kind of differences? Please mention as many as you can.
(2) Did you think any parts of any of the systems were controlled by a person? If yes, which parts did you think were controlled by a person?
(3) Please add any general comments about your experience talking to the two systems

## 5 RESULTS

### 5.1 Questionnaire Results

For evaluating the differences in perception between the two systems, we took a mean score for each participant for each question domain: engagement, enjoyment, and social presence (see Fig. 7). Mean engagement dashboard = 2.8 (std = 1.05), mean engagement VR = 3.38 (std = 1.08). Mean enjoyment dashboard = 3.5 (std = 0.83), mean enjoyment VR = 4.02 (std = 0.84). Mean social presence dashboard = 2.16 (std = 0.3), mean social presence VR = 2.6 (std = 0.73).

Direct comparison of the two systems using a two-sample t-test showed a significant difference in enjoyment scores (t = -2.6382, df = 8.0214, p-value = 0.02973) and social presence scores (t = -2.5136, df = 6.4132, p-value = 0.04319). For engagement scores, the difference was not significant (p-value > 0.05).

For the robot features, we compared the mean score for each feature by condition. We found that on average, all features were rated higher in the VR condition, however, the difference in scores varied for each question (Fig. 6). This way, facial expressions and head and eye movements had a minimal difference in score by condition, while a higher difference was found for voice, speech content, and reactions to the user's utterances.

*5.1.1 Engagement.* The questionnaire results showed that the two WoZ systems provided similar engagement in the interaction between the robot and the participant with a tendency towards the VR system being more engaging. This result can be due to the small sample size of the pilot study (N=6), thus further exploration is needed to investigate the effect of different WoZ systems on conversational engagement. However, one of the participants reported that they felt a strong difference between the systems when asked for general comments about the experience.
Participant 4: *"The first [VR] was a lot more engaging, maybe because of how surprised [I was] at how good it was. The second system [Dashboard] was trickier to engage with."*

*5.1.2 Enjoyment.* A significant difference in perceived enjoyment of the interaction was found between the systems showing that the VR system was perceived as more enjoyable to talk to.
Participant 5: *"The second system [VR] had more soul to it, it had jokes and it felt like it maintained the context of the dialogue better."*

*5.1.3 Social presence.* The VR WoZ system had a significantly higher score in social presence indicating that the participants felt more like they were interacting with a social being in the VR condition compared to the dashboard. However, it is worth noting

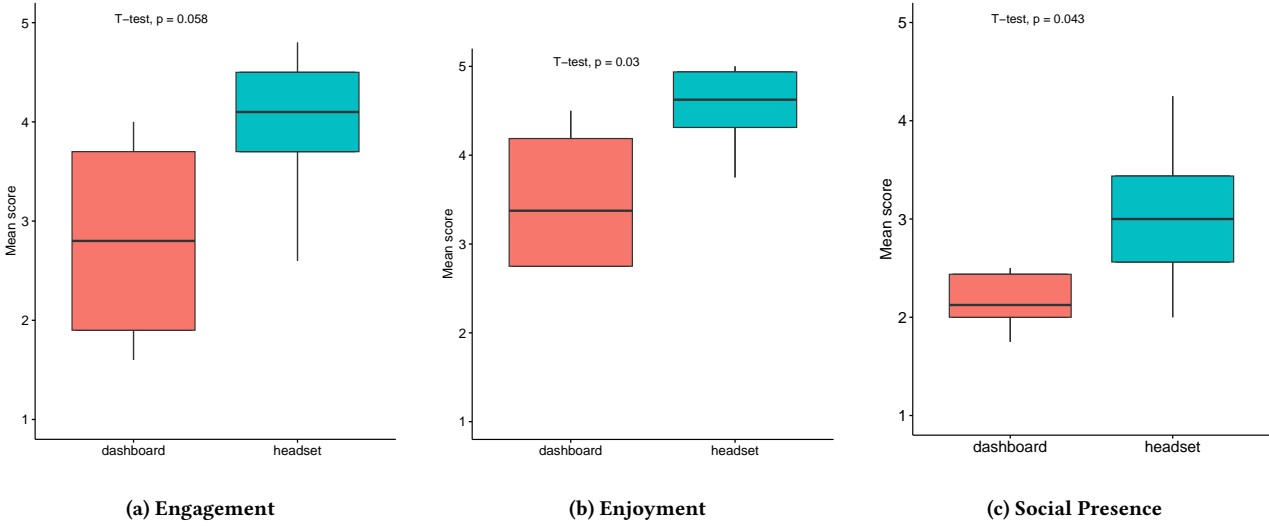

(a) Engagement          (b) Enjoyment          (c) Social Presence

**Figure 5: Mean interaction questionnaire scores by condition**

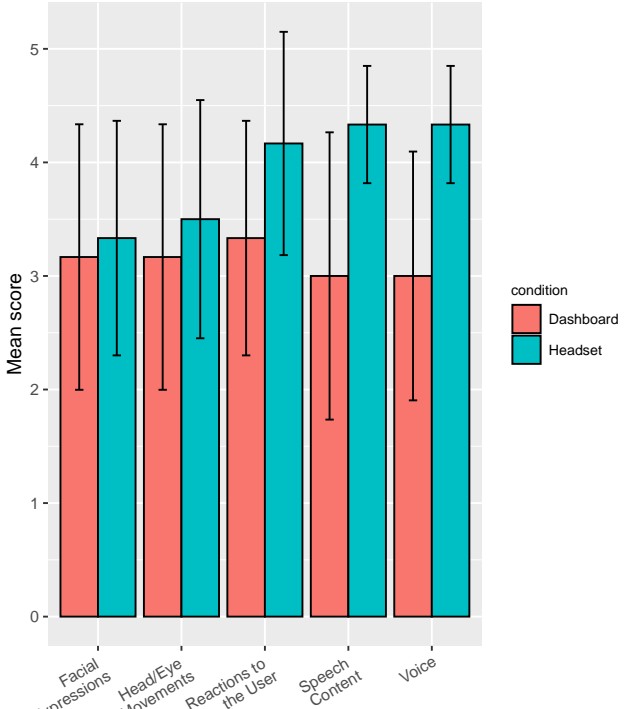

**Figure 6: Mean scores for the robot's features per question by condition. Error bars indicate standard deviation.**

that in both conditions social presence resulted in lower mean scores than engagement and enjoyment.

Participant 1: *"Second system [VR] felt more interesting to talk to,* *more engaged in the conversation, but overall both seemed really close to talking to the actual person."*

*5.1.4 Robot's Features.* Several participants noted the differences in the robot's voice between conditions. The participants that commented on the voice agreed that the VR system (which had the natural voice of the operator) was preferable.

Participant 2: *"The tone of the second system [VR] was more engaging, the first system [Dashboard] sounded very flat."*

Participant 4: *"The voice in system 2 [Dashboard] was a lot more robotic and more difficult to engage with"*

The head movements and some of the facial expressions were noted to be different between the systems. However, there was no consistent pattern in preference between the systems.

Participant 3: *"The movement of the eyebrows for the second system [Dashboard] did not always feel to match the discussion."*

Participant 2: *"At one point the first system [Dashboard] frowned seemingly for no reason."*

Participant 5: *"I think that the second system [VR] smiled more and sometimes tilted its head that gave a more realistic feeling overall, but the eye movement of the first system [Dashboard] felt a little bit more real than of the second system [VR]."*

Participant 6: *"During the first conversation [VR] the head of the robot moved and it was fun"* (the operator made a joke about the robot practicing nodding)

In addition, some of the participants noted the difference in timing of the robot's responses and described the VR system as having better timing in its responses.

Participant 2: *"The first one [Dashboard] felt very choppy to talk to, it interrupted me a lot. The conversation with the second [VR] was much smoother and nicer"*

Participant 3: *"The first system [VR] was generally more responsive <...> especially when I asked the questions. The second system [Dashboard] did more take its time."*

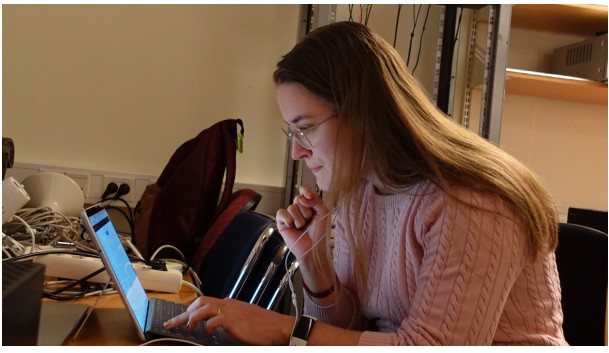

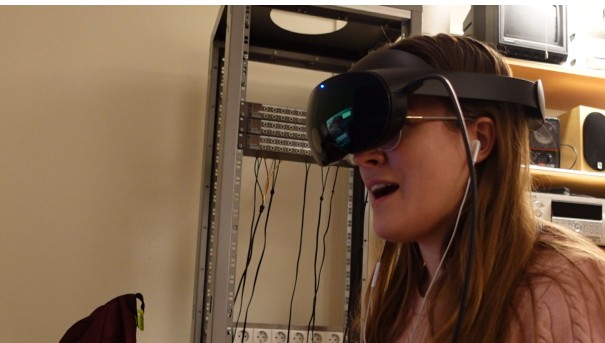

**(a) Operator using a dashboard WoZ interface where keystrokes on a laptop are used to control the robot.**

**(b) Operator using VR headset to control the robot.**

**Figure 7: The operator using two WoZ systems**

Participant 6: *"Second system [Dashboard] was slower. Sometimes it took bigger pause than it was needed."*

*5.1.5 Perceived Autonomy.* Five participants who reasoned about the robot's autonomy assumed that the robot had a lot of autonomy in both conditions. One participant stated that they did not think about whether the robot was controlled by a human or not. This indicates that both systems can be perceived as autonomous robot behavior. However, it is noteworthy that 5 out of 6 participants in the study had not previously experienced interacting with social robots. We assume that this result could differ if the participants possessed prior familiarity with social robotics.

## 6 DISCUSSION

We conducted a pilot study comparing a dashboard- and a virtual reality (VR)-based Wizard-of-Oz (WoZ) system for a human-robot conversational task. Our aim was to ensure that the interactions between the conditions were as close as possible, allowing for the freedom of choice of responses necessary for naturalistic dialogue. To facilitate this, we opted for a single operator to test both systems, with the assumption that a naturalistic setting for a user study would involve interacting with multiple participants throughout the day.

Our findings showed that, overall, the VR system was perceived better by the participants in terms of enjoyment and social presence. This was evidenced by higher scores on the Likert scale questions and in the majority of the open-ended reflections. Furthermore, while we did not find a significant difference between the conditions in terms of engagement questions, the open-ended reflections indicated a tendency toward VR being more engaging. Consequently, the effect of the teleoperation method on conversational engagement warrants further investigation.

Utilizing VR-mediated WoZ systems offers the potential to harvest richer data in terms of facial expressions and eye movements, which are of particular significance for the robot employed in this experiment. With the advancement of technology, which is increasingly able to harness vast amounts of data to produce both artwork and human-like conversational behavior, we find ourselves on the brink of achieving a level of social behavior indistinguishable from

that of humans. This development could potentially lead to the creation of an "uncanny valley," yet none of the participants in this experiment reported feeling any sense of creepiness or discomfort.

Notably, the participants believed in the robot's autonomy in both conditions despite the operator's natural voice being played in the VR condition. While this result can be affected by the participants not having interacted with social robots before, to make the robot more believable, the operator's voice can be passed through a filter more resembling state-of-the-art TTS voice synthesizers. However, one of the participants noted in general comments that the voice in the VR condition resembled a synthesized voice, which could be due to the quality of the speaker used for voice output. Another possible explanation for why the participants believed that the robot was autonomous can be the media coverage of recent developments in AI technology and the belief that the field is quickly developing.

As expected, several participants noted the dashboard system being worse in the timing of the responses: either interrupting the participant or taking too much time to answer. This is in line with the operator reporting on conversational failures during the experiments in the two conditions. The conversational data from our experiments will be further analyzed by examining the conversation dynamics in each condition as it is an important part of the functionality of the systems. We will evaluate the number of pauses, overlaps, transitional gaps, number of turns taken, and turn duration.

When evaluating the robot's features manipulated between the conditions, the VR condition was also rated higher overall. The highest difference was obtained for voice, speech content, and the reactions to the user. However, the mean scores for the robot's facial expressions and head and eye movement had a minimal variation between the conditions. This result can be explained by the content of the dialogue, which was designed for second language practice. As the dashboard system had a predefined dialogue tree, the operator structured the dialogue script in the VR condition based on the dialogue tree. The content of the dialogue required minimum eye and head movements, thus the conditions could be perceived similarly. It would be interesting to evaluate the VR system in a

multi-party setting requiring the robot to switch attention between the participants; or in a task requiring mutual visual grounding.

Surprisingly, facial expressions were also rated with a minimal increase in the VR condition, however, several participants pointed out that the robot smiled more in the VR condition. By reviewing the footage from the experiment, we have concluded that in the Virtual Reality condition, there were some eye movement inaccuracies in Furhat's gaze. Specifically, the eyes were sometimes partly closed which made it look like the robot had heavy eyelids. We will further assess if these were created by the wizard operator in this study wearing glasses, a problem in our system, or a misaligned placement of the headset while switching between WoZ systems. Another reason can be the sensitivity to mild expressions: the operator maintained a neutral posture without expressing exaggerated emotions, as corresponded to the content of the dialogue.

As mentioned previously, the dashboard setup used in the current study has some aspects of semi-autonomy, which significantly reduces the cognitive load on the operator. However, a more typical dashboard setting would involve all response options present on the dashboard. In the future, we aim at comparing the systems at three levels of conversational autonomy: simple dashboard, semi-autonomous dashboard, and fully controlled VR setup. In future work, this system will also be further assessed from the operator's point of view. To the best of our knowledge, no direct comparison between typical screen-based and VR-mediated WoZ systems in terms of the operator's perception has been conducted. The evaluation will focus on measuring general usability, the operator's cognitive and physical load, and perceived social presence.

## 7 CONCLUSION

This study has demonstrated the advantages of utilizing VR-mediated WoZ interfaces as an accessible and flexible method for testing human-robot interactions. The research has demonstrated that recent advances in AI technology have enabled a richer and more engaging experience for users, without compromising the autonomy of the robot or causing an unsettling sense of the uncanny valley. The VR system was found to be more enjoyable to interact with, and it exhibited a higher level of social presence. The results thus suggest that VR-mediated WoZ interfaces have a promising future in the field of HRI.

## ACKNOWLEDGMENTS

We would like to thank Ronald Cumbal for providing access to the semi-autonomous wizard operating system. This work was supported by the Digital Futures project "Using Neuroimaging Data for Exploring Conversational Engagement in Human-Robot Interaction" and the Swedish Research Council project 2021-05803.

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
