# OpenReview forum: "Comparing Dashboard and Virtual Reality Wizard-of-Oz Setups In a Human-Robot Conversational Task"
_humanrobotinteraction.org/HRI/2023/Workshop/VAM-HRI — VAM-HRI 2023 Oral_

### Official Review · Program_Chairs · 2023-02-24
**Accept**

**Rating:** 8
**Confidence:** 5

**Review:**

Reviewer 1: This paper explores the differences between a traditional social robot WoZ set up (i.e., desktop dashboard) vs a VR system (i.e., facial tracking) validated via a pilot study (n=6, operator=1) with the goal of the VR system to be at least as good as, if not better than, the traditional set up. They report participants having a more engaged/higher social presence with no significant other differences found between the interfaces.


I recommend this paper is accepted and would make a great contribution to the workshop! The interface is quite novel and expands discussion into more social components of VAM.


Nitpicks and comments:
* This is quite a unique way to control robot faces (at least from my knowledge) and quite cool to see
* Please add the number of participants within the abstract
* For background, there are quite a few comparisons (I believe) between different teleop modes (largely VR vs desktop). Now these are for different teleoped robots (largely arms and mobile robots) and not necessarily social/face-based and for WoZ. Please see the following from an initial google scholar search of “vr teleoperation robot vs desktop”; I think you should include this as a reference and mention WoZ/social as the main factor missing:
   * https://link.springer.com/chapter/10.1007/978-3-030-28619-4_28
   * https://ieeexplore.ieee.org/abstract/document/8968598?casa_token=u7Mlp5rT_6UAAAAA:-sUzRVeow7UrbCcWouvhpv7t7ue0UiNHgXeUv5szZwpgVsvF8-M3AgqFL7kud9ufydPjF3LK
   * https://www.tandfonline.com/doi/abs/10.1080/10447318.2015.1039909
* Please report how participants were recruited (https://ieeexplore.ieee.org/document/9900744 , https://zhaohanphd.com/publications/hri23lbr-towards-improved-replicability-of-human-studies-in-human-robot-interaction-recommendations-for-formalized-reporting/ )
* For the operator, please expand on what you mean by “good understanding”, did they create the interface/part of the development/how many hours have they used it vs dashboard
* Study outline is quite good and including the full set of questionnaires is very appreciated
* You will want to thoroughly address the limitation of the small sample size and that it is a possible helpful indicator for future work directions but in no way definitive. This goes both ways, for “positive” or “negative/non” results. E.g., 5.1.1: “This result can be due to the small sample size of the pilot study (N=6), thus further exploration is needed to investigate the effect of different WoZ systems on conversational engagement” -> this implies small sample sizes are the reason for a result which cannot be generalized/true. Small sample sizes are helpful (I’m a big supporter of pilot study papers) but you can’t really have it both ways for good and bad results as the “scientific” argument is “you should have had a bigger sample then”. This also alludes to reporting normal distribution stats (e.g., mean and standard deviation). I think there is a really great read on HRI Likert usage that I find additionally helpful: https://arxiv.org/abs/2001.03231
* Thus the highlight of the paper “should*” be the qualitative and behavioral data which I greatly appreciate you putting in the quotes

Review 2:
This paper explores the use of VR for wizard-of-oz setups with social robots, comparing against traditional interfaces like a dashboard setup. Overall, I recommend accepting this paper to the workshop, it is relevant to the community and will be interesting for people to discuss and understand how VAM technologies can be used to make engaging social interactions with robots.

Feedback:
- In the VR condition, the operator’s voice was played through a speaker, whereas in the dashboard, the robot’s voice was played instead. What is the motivation for this distinction, why not let the dashboard operator use their voice and read the options from the dashboard?
- Although this study was done with a single operator (author of the paper), it would be interesting to evaluate how non-experts find using the system themselves, to understand how VR can be used to make operation easier.
- In the introduction, one of the citations is a ?

---

### Decision · Program_Chairs · 2023-03-02

Accept (Oral)